# Cholestatic HCV Cryoglobulinemia: A New Clinical and Pathological Entity before and after Direct-Acting Antiviral Therapies—A Case-Control Study

**DOI:** 10.3390/ijms25020784

**Published:** 2024-01-08

**Authors:** Serena Ammendola, Sara Romeo, Filippo Cattazzo, Anna Mantovani, Donatella Ieluzzi, Veronica Paon, Martina Montagnana, Sara Pecori, Anna Tomezzoli, Andrea Dalbeni, David Sacerdoti

**Affiliations:** 1Division of Pathology, Department of Diagnostics and Public Health, Azienda Ospedaliera Universitaria Integrata of Verona, 37134 Verona, Italy; sara.pecori@aovr.veneto.it (S.P.); annatomez@hotmail.com (A.T.); 2Division of General Medicine C, Department of Medicine, Azienda Ospedaliera Universitaria Integrata of Verona, 37134 Verona, Italy; sara.romeo26@yahoo.it (S.R.); cattazzo.f@gmail.com (F.C.); anna.mantovani@aovr.veneto.it (A.M.); donatella.ieluzzi@aovr.veneto.it (D.I.); veronica.paon@aovr.veneto.it (V.P.); andrea.dalbeni@aovr.veneto.it (A.D.); 3Liver Unit, Department of Medicine, University of Verona, 37129 Verona, Italy; david.sacerdoti@univr.it; 4Section of Clinical Biochemistry, Department of Neurological, Biomedical and Movement Sciences, University of Verona, 37129 Verona, Italy; martina.montagnana@univr.it

**Keywords:** cryoglobulinemia, HCV, DAAs, cholestasis, HCV RNA, serum biomarkers, liver biopsy, inflammatory infiltrates, histology, vasculitis

## Abstract

Twenty-nine patients with HCV infection (HCV+) and mixed cryoglobulinemia (MC+) were retrospectively selected and matched for age and sex with 31 HCV+ MC− patients. Biomarkers of cholestasis (direct bilirubin, alkaline phosphatase, and gamma-glutamyl transferase), HCV-RNA and genotype, and plasma cryoprecipitates were measured before and after virus eradication; liver histology and plasma cells (aggregation and distribution), observed blinded by two pathologists, were analyzed. Sixty participants (mean age: 56.5; range: 35–77, males: 50%) with HCV infection were enrolled. Cholestasis (≥2 pathologically increased cholestasis biomarkers) was significantly higher in the MC group (*p* = 0.02) and correlated with cryoglobulinemia (OR 6.52; *p* = 0.02). At liver histological assessment, plasma cells were significantly increased in the MC+ group (*p* = 0.004) and tended to form aggregates more than the control group (*p* = 0.05). At multivariate analysis with MC, age, HCV-RNA, HBV diabetes, and cirrhosis, cholestasis was only significantly correlated to MC (OR 8.30; *p* < 0.05). In 25% patients, MC persisted after virus eradication with new antiviral treatment. Our study identified for the first time an association between MC, cholestasis, and an increased number of intrahepatic plasma cells in chronic hepatitis C (CHC) patients before virus eradication. Future studies are required to understand how MC contributes to liver damage and how its persistence affects the patients’ follow-up after antiviral therapies.

## 1. Introduction

Mixed cryoglobulinemia (MC), comprising type II and III cryoglobulinemia according to Bouet’s classification [1], is defined as the presence of circulating plasma monoclonal and/or polyclonal immunoglobulins (Igs) that precipitate at less than 37 °C, consequently producing organ damage through an immune-mediated mechanism [1,2].

The classical clinical manifestation of type II and type III cryoglobulinemia [3] is represented by the triad of purpura, weakness, and arthralgia [4] frequently burdened by multiple visceral complications [5]. Cryoglobulinemia caused by hepatitis C virus (HCV) infection is typically a mixed form determined by a B-cell hyperactivation directly mediated by HCV via the cell surface protein CD81 [6].

Since the first findings of the presence of anti-HCV antibodies and of HCV-RNA sequences in cryoprecipitates [7], the pathogenesis of mixed cryoglobulinemia has been extensively investigated and MC can be regarded as a direct consequence of HCV infection in up to 80–90% of cases [2,5,8]. Cryoglobulinemic vasculitis is a systemic disorder, determined by the deposition of cryoglobulins within vessel walls associated with increased mortality and high healthcare costs in patients with chronic hepatitis C (CHC) [9], and HCV eradication is not always followed by a resolution of the damage caused by MC. Indeed, as documented for other extrahepatic manifestations of CHC, such as lymphoproliferative disorders [10,11], renal and ocular involvement, and cardiovascular and cerebrovascular disorders [10,12], after clearance of HCV via antiviral therapy, in particular with Direct-Acting Antiviral (DAA) therapies, disappearance of cryoglobulinemia is observed in only about 50% of cases [10], which highlights how the aforementioned therapy does not have consistent effects on the presence of MC. A correlation between HCV infection and cholestasis has already been established [13,14]; indeed, the onset of cholestasis in CHC is thought to be determined by direct viral cytotoxicity of hepatocytes and is associated with a weak HCV-specific T-cell response, which is predominantly carried out by Th2 cells, leading to increasingly higher levels of viral replication and amplification of the intrahepatic damage [15].

On the other hand, data on how cryoprecipitates interact with the liver parenchyma are scarce and their potential role in developing and sustaining cholestasis is still unexplored.

Based on these premises, this study aims to investigate the correlation between MC and intra-hepatic cholestasis in chronic hepatitis C (CHC) patients before and after HCV eradication, particularly in patients treated with DAAs.

## 2. Results

Patients with CHC diagnosed at the Verona University Hospital between 2000 and 2017 with non-eradicated HCV infection, histologically proven chronic hepatitis, and concurrent MC were identified through clinical records and selected as the study group. A group of age-, sex- and HCV genotype-matched patients with non-eradicated HCV infection and histologically proven chronic hepatitis who did not develop MC was also selected as the control group.

Sixty patients, age, sex, and HCV-genotype matched, with non-eradicated HCV, were enrolled: 29 within the study group (MC+) and 31 in the control group (MC−). All patients were treated with interferon and ribavirin between 2010 and 2017 (8.1%) or DAAs after 2017, and all achieved a sustained virological response (SVR). The general characteristics of the study population are described in Table 1.

In brief, the mean age was 56.5 (35–77) years and 50% were male, HCV genotype was 1 in 61% of cases, 2 in 18.6% of cases, 3 in 15.3% cases, and 4 in 5.1% cases; 17 (28.3%) patients had a concurrent hepatitis B virus (HBV) infection. Cholestasis was observed in 11 (18.3%) cases. At univariate analysis, cholestasis was significantly correlated with the presence of cryoglobulinemia (OR 6.52 CI 95%: 1.27–33.45; *p* = 0.02).

At multivariate analysis (Table 2), the presence of MC+ was the only feature significantly correlated to cholestasis (OR 8.30 CI 95%: 1.16–61.82; *p* < 0.05), while age, HCV RNA, diabetes, and cirrhosis were not significantly associated.

Thirty-four patients (55.8%) received a histological diagnosis of cirrhosis without significant differences in distribution between the two groups (*p* = 0.06) and 25 (41.4%) showed micro- or macrovesicular steatosis. Plasma cells were completely absent in 3 (5.4%) cases and present in the remaining 57 (94.6%). Among the cases in which plasma cells were detected both morphologically and immunohistochemically, aggregates were found in 21 (35.7%) cases. Plasma cells on liver histology were found in a significantly higher number in the MC+ group (*p* = 0.004) (Figure 1).

### Cholestasis Biomarkers and MC after HCV Eradication

Cryoglobulinemia was present in 54% of patients from the MC+ group even after HCV eradication, and cholestasis was still present despite virus eradication in 25% of patients, with no significant variations in cholestasis biomarker levels among the study and control groups (*p* = 0.65) (Table 3).

## 3. Discussion

The most relevant findings of this study can be summarized as follows: (i) in both univariate and stepwise multivariate analysis, cholestasis was significantly correlated with cryoglobulinemia; (ii) the presence of a higher number of plasma cells in the liver parenchyma was significantly more frequent in patients with MC+ compared with MC− cases; and (iii) virus eradication was not associated with the disappearance of serum cryoglobulins in a large percentage of patients.

Several studies linked HCV infection to a high risk of developing intrahepatic cholestasis [14,16,17], in particular during pregnancy [17] and even after solid organ transplantation [18], although, cholestasis can develop in immunocompetent individuals or be determined by different types of non hepatotropic viruses, intrahepatic cholestasis can be determined by different types of non-hepatotropic viruses, including Epstein–Barr (EBV) [19], HIV, cytomegalovirus (CMV) [20], or pathogens such as cryptosporidia and microsporidia [21]. Mixed cryoglobulinemia is most commonly caused by HCV infection, and affects 30% to 50% of patients with CHC [22]; in these patients, cryoglobulinemia is mainly asymptomatic, but 5 to 10% of patients with HCV-associated cryoglobulinemia will experience a more severe, potentially life-threatening, clinical course with the onset of cryoglobulinemic vasculitis [23].

Although the causative role of viral infections in the development of intrahepatic cholestasis has been documented [14,16,22], according to our findings, a significant increase in the levels of biomarkers of cholestasis can be observed in patients who develop MC during the natural history of the untreated HCV infection. However, a systematic appraisal of the relationship between cholestasis and the presence of cryoprecipitates in CHC patients was lacking. To the best of our knowledge, this is the first study in which a significant increase in intrahepatic cholestasis indices has been documented in patients with HCV-related cryoglobulinemia compared with those without serum cryoglobulinemia.

Indeed, the present study shows a strict relationship between cholestasis and MC, and between MC and the presence of an increased number of plasma cells in the hepatic inflammatory infiltrate, independently from differences in the viral load, cirrhosis, and other comorbidities. Noticeably, in our experience, serum biomarkers were more accurate in determining the presence of cholestasis compared with the histological assessment of liver tissue. Indeed, the most reliable histological feature of cholestasis is the presence of yellow–brown bile pigment in the perivenular spaces, in hepatocytes, or within bile ducts [24]. However, in our cohort, histological signs of cholestasis were mostly indirect, such as the presence of feathery degeneration of periportal hepatocytes, showing enlarged, swollen cells with clear, finely granular cytoplasm, the presence of ductular proliferation, but also brown pigmentation of the hepatocytes, which was visible in most affected cases. Nonetheless, most of the samples analyzed were biopsies; therefore, the assessment could have been limited by the relatively small amount of tissue available for evaluation. Moreover, the relative weight on the onset of cholestasis determined by cryoglobulinemia and by plasma cells in liver tissue is yet to be determined. Despite further studies being required to address the precise mechanism involved in the increase in cholestasis in MC+ patients, based on the findings of this study, it can be hypothesized that the presence of immune complexes may determine not only a vasculitic type of injury, which can persist after virus eradication [25] but also direct damage to the hepatic parenchyma. Indeed, as documented in the kidney, where circulating immunocomplexes and cryoglobulins deposit in glomerular structures and in the subendothelial and subepithelial structures [26,27], we could hypothesize that a similar mechanism of extravascular damage could involve the portal spaces and hepatocytes, especially since the impaired liver clearance capacity increases deposition and persistence of cryoglobulins [28]. Another important consideration regards the role of immunohistochemistry in evaluating plasma cell content in liver biopsies from HCV patients with MC. This ancillary technique increased the accuracy of the histological evaluation via H&E, especially in cases with only a few plasma cells, which can be difficult to detect and count within a lymphohistiocytic inflammatory infiltrate. Moreover, immunohistochemistry was also useful to better define cases with plasma cell aggregates. Our study presents some limitations, in particular, it is a retrospective study where all the histological material was obtained before virus eradication, and liver biopsies were not repeated after achieving a sustained virological response; moreover, follow-up bioumoral data for every patient were not taken at a fixed time interval; therefore, to conduct the analysis on the presence of cholestasis after DAAs therapies, the most recent value obtained in the 3–5 year follow-up period was selected. Finally, the minimum number of plasma cells required to form an aggregate was arbitrarily chosen due to the lack of a reference cut-off value in the literature [29]. Due to the retrospective nature of the study, liver tissue was not available for further analysis via immunofluorescence and electron microscopy to characterize the presence and localization of cryoprecipitates within the hepatic tissue. These techniques could indeed aid in the characterization and interpretation of the mechanisms linking MC and cholestasis in CHC patients.

Nonetheless, we are the first to describe the presence of a more consistent plasma cell content in liver biopsies and an increase in biochemical markers of cholestasis in patients with CHC-related cryoglobulinemia. These observations, if confirmed by larger studies, could suggest testing for MC in those patients with increased plasma cell contents in liver biopsy and vice versa; a two-fold or higher increase in biomarkers of cholestasis could indicate the development or recurrence of MC in patients with active disease or after eradication. In fact, despite DAAs revolutionizing the therapeutic approach to HCV infections, the possible persistence of MC even after HCV eradication should be taken into consideration by the clinician [25] since these patients could require additional treatments for symptoms of MC.

In conclusion, this study identified for the first time a particular subset of CHC patients with cryoglobulinemia who developed cholestasis, probably through an immune-mediated mechanism. The combination of cholestasis and cryoglobulinemia could define a new entity in the spectrum of CHC. Further studies are required to confirm these findings and to understand the pathogenesis of cholestatic HCV-related cryoglobulinemia despite virus eradication.

## 4. Materials and Methods

The study protocol was approved by the Institutional Ethics Committee of Verona (Italy) and all patients provided written informed consent to be included in the study (2730CESC-VR).

For the whole population, data on cholestasis parameters including direct bilirubin, alkaline phosphatase (APL), gamma-glutamyl transferase (GGT), HCV-RNA serum levels, and HCV genotype were retrieved. Serum levels of alanine aminotransferase (ALT) and aspartate aminotransferase (AST) were also recorded. All patients underwent liver biopsy before HCV eradication with DAAs.

The same bioumoral data were also collected after HCV eradication (between the 3- and 5-year follow-up examinations). Formalin-fixed, paraffin-embedded (FFPE) tissue and hematoxylin–eosin (H&E)-stained glass slides of the liver biopsy were also retrieved for histological revision and immunohistochemical analysis. Patients with autoimmune disease, hematological malignancies, concurrent HCV/HIV infection, and cases in which FFPE material was unavailable for immunohistochemical analysis were excluded.

### 4.1. Cryoglobulin Detection Methods

Samples were collected in a 10 mL tube without anticoagulant and without separator gel, and then preheated at 37 °C (Vacutest, Kima, Padova, Italy) [30].

After withdrawal, the tube was inserted into a portable and preheated device (at 37 °C) [31] and transported to the laboratory, following at least 1 h after withdrawal to allow for clotting. Centrifugation was performed at 37 °C at 1500× *g* for 15 min. The serum was divided into two tubes and stored at 4 °C for 7 days [32]. Once a cryoprecipitate (CPT) formed, one tube was returned to 37 °C to verify the dissolution of the CPT [33]. Particulate, lipemic, as well as hemolyzed or strongly icteric samples were discarded.

Cholestasis was defined as an increase greater than twice the upper normal limits of at least two bioumoral parameters among conjugated bilirubin, GGT, ALP, and bile acids, lasting more than 6 months [34].

### 4.2. Pathological Assessment

All histological glass slides were revised by two pathologists for histological signs of chronic C hepatitis and/or cirrhosis. Histological grading and staging of chronic hepatitis were assessed according to Ishak et al. [35]. Moreover, the presence of inflammatory infiltrates and the presence, localization, and distribution of plasma cells within the hepatic tissue were also recorded. The assessment was undertaken by the study investigators blinded to the assignment of the biopsies to either the study- or the control group.

### 4.3. Immunohistochemistry

To highlight the number and distribution of plasma cells, all cases from the study and the control groups were immunostained using an anti-CD38 antibody (clone SP149; Ventana Benchmark Ultra, Ventana Medical Systems, Tucson, AZ, USA) by means of an automated immunostainer (Leica Biosystems, Newcastle, UK).

Plasma cell count on the immunohistochemically stained sections was conducted manually in one “hotspot” (i.e., the microscopic field, measuring 0.229 mm^2^, with the highest number of plasma cells).

The presence or absence of plasma cells, the plasma cell count in one hotspot, and their distribution within the hepatic parenchyma (plasma cells confined to the portal spaces and/or within the hepatic lobule) were recorded. Finally, the presence of plasma cell aggregates, defined as a cluster of 5 or more plasma cells, was annotated.

### 4.4. Statistical Analyses

Continuous variables were presented as mean ± standard deviation for normally distributed data or as median with interquartile range if a non-normal distribution was found through the Shapiro–Wilk test. Categorical variables were expressed as percentages. Independent samples from either the Student *t*-test or the Mann–Whitney U test were used to compare continuous variables according to the data distribution (normal or non-normal). Categorical variables were compared using the Chi-square test; the Fisher exact test was performed for non-normally distributed data.

Multivariate analysis was conducted using binomial logistic regression to find significant correlations between the increase in cholestatic parameters > 2 times the upper limits of normal values and the presence of cryoglobulinemia, the number of plasma cells in the hotspot, and the presence of plasma cell aggregates and other clinical pathological parameters. All tests were 2-sided, and *p*-values < 0.05 were considered statistically significant.

Statistical analyses were performed using jamovi (The jamovi project (2022). jamovi (Version 2.3). Retrieved from https://www.jamovi.org, accessed on 13 August 2023).

## Figures and Tables

**Figure 1 ijms-25-00784-f001:**
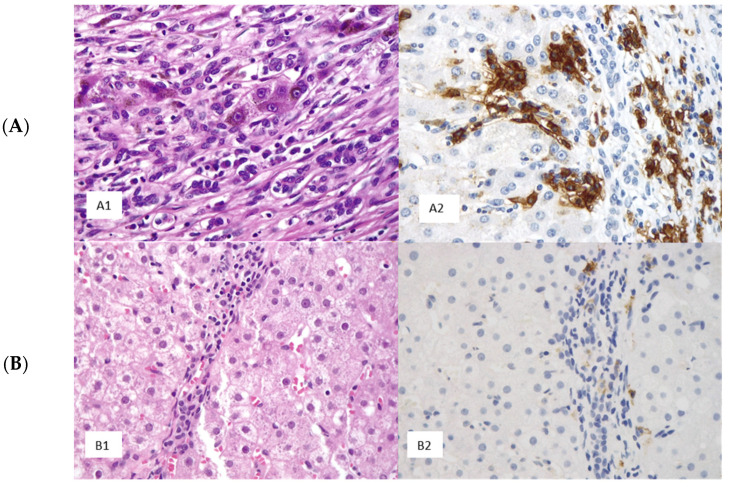
Different contents of plasma cells in hepatic biopsies from HCV-positive patients with (**A**) and without (**B**) cryoglobulinemia. A case with numerous plasma cells detected in hematoxylin and eosin (H&E, 400×) (**A1**) and via immunohistochemistry (IHC, 400×) with anti-CD138 antibody (**A2**). A case with isolated plasma cells detected in H&E, 400× (**B1**) and IHC, 400× (**B2**).

**Table 1 ijms-25-00784-t001:** Main clinical and pathological characteristics of the study and control groups.

	Whole Population n = 60	MC− n = 31	MC+ n = 29	*p*
Age (years)	56.5 (35–77)	57 (35–77)	56 (35–77)	0.35
Sex (M) (%)	50	54.8	44.8	0.60
Aetiology				
Steatosis (%)	41.4	41.9	37.9	0.79
Cirrhosis (%)	55.8	41.9	55.2	0.06
Diabetes (%)	18.4	16.1	13.8	0.70
Current/previous alcohol consumption (%)	20	12.9	27.6	0.16
Previous antiviral therapy (%)	8.3	9.7	6.9	0.70
HBV+ (%)	28.3	19.3	37.9	0.11
Genotype				0.55
1 (%)	61	67.7	51.7	
2 (%)	18.6	16.1	20.7	
3 (%)	15.3	9.7	20.7	
4 (%)	5.1	6.4	3.4	
HCV-RNA (copies/mL)	1.34 × 10^6^ ± 1.99 × 10^6^	1.60 × 10^6^ ± 2.20 × 10^6^	1.06 × 10^6^ ± 1.75 × 10^6^	0.06
Biochemical parameters				
AST (UI)	60.5 (15–355)	53 (20–332)	64 (15–355)	0.16
ALT (UI)	68.5 (16–369)	68 (22–315)	81 (16–369)	0.81
GGT (UI)	50.5 (3–1189)	69 (11–605)	44 (3–1189)	0.31
ALP (UI)	80.5 (1–376)	82 (38–171)	78 (1–376)	0.92
Direct bilirubin (mg/dL)	0.2 (0.1–6.6)	0.2 (0.1–0.01)	0.2 (0.1–6.6)	0.63
Cholestasis > 2x UNL (%)	18.3	6.4	31	0.02
Histopathology				
Presence of plasma cells (%)	94.6	92.8	93.1	1
N/hotspot 40x	26.1 ± 20.6	18.5 ± 16.7	33.6 ± 21.5	<0.05
Plasma cells location (Portal space *, lobule **, both ***) (%)	* 64.3	* 67.8	* 58.6	0.66
	** 5.4	** 7.1	** 3.4	
	*** 30.4	*** 25	*** 34.5	
Plasma cell aggregates (%)	35.7	21.4	48.3	0.05

MC: mixed cryoglobulins; HCV: hepatitis C virus; HBV: hepatitis B virus; HIV: human immunodeficiency virus; HCV-RNA: hepatitis C virus-ribonucleic acid; AST: aspartate aminotransferase; ALT: alanine aminotransferase; GGT: gamma-glutamyl transferase; ALP: alkaline phosphatase; UNL: upper normal limit; *: plasma cells only in portal spaces; **: plasma cells only in lobules; ***: plasma cells in both portal spaces and lobules.

**Table 2 ijms-25-00784-t002:** Univariate and multivariate logistic regression.

Univariate Analysis			
	OR	95% CI	*p*
MC+	6.52	1.27–33.45	0.02
Age	1.05	−0.01–0.12	0.09
Sex	0.50	−2.03–0.67	0.32
HCV-RNA	1.00	−8.30 × 10^−7^–1.98 × 10^−7^	0.22
HBV	1.58	0.39–6.30	0.52
Diabetes	2.35	−0.75–2.46	0.29
Steatosis	1.22	−1.12–1.53	0.76
Cirrhosis	4.72	0.91–24.60	0.06
Hematologic diseases	1.33	−2.09–2.66	0.81
Presence of plasma cells	0.46	−3.26–1.73	0.54
N° plasma cells/hotspot 40X	1.05	0.01–0.08	0.004
Plasma cell aggregates	2.65	−0.36–2.32	0.15
Multivariate analysis			
	OR	95% CI	*p*
MC+	8.30	1.16–61.82	<0.05
Age	1.1	0.95–1.16	0.31
HCV-RNA	1.00	1.00–1.00	0.95
Diabetes	2.02	0.27–14.84	0.49
Cirrhosis	1.50	0.20-11.14	0.68

**Table 3 ijms-25-00784-t003:** Cholestasis before and after HCV eradication in the MC+ group.

	MC+ Group		
	Before HCV Eradication	After HCV Eradication	*p*
GGT (UI)	45 (3–1189)	61.5 (18–270)	0.025
ALP (UI)	78 (1–376)	133 (91–234)	0.010
Direct bilirubin (mg/dL)	0.2 (0.1–6.6)	0.7 (0.3–0.7)	0.001
Cholestasis > 2x UNL (%)	32.2	25	0.65

GGT: gamma-glutamyl transferase; ALP: alkaline phosphatase; and UNL: upper normal limit.

## Data Availability

Data are available upon reasonable request to the corresponding author.

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
