# Peer review of "Cholestatic HCV Cryoglobulinemia: A New Clinical and Pathological Entity before and after Direct-Acting Antiviral Therapies—A Case-Control Study"

_ijms, 2024, doi:10.3390/ijms25020784_

Round 1

Reviewer 1 Report

Comments and Suggestions for Authors

Overall an interesting paper, however I have a few issues that I think they need to be adressed.

1. In introduction section, it should be mentioned that cryoglobulinemia is only one of the extrahepatic manifestations of HCV and that DAAs seem to have incosistent effects in them (ref: Kuna L, J Clin Transl Hepatol. 2019; Pol S, Nat Rev Gastroenterol Hepatol. 2018; Androutsakos T, Viruses 2023; Mazzaro C, Viruses 2021 among others)

2. The 3 patients with HIV co-infection should be excluded, since both cholestasis and plasma cell infiltrates could be attributed to HIV infection per se (ref: Te HS, Clin Liv Dis 2004)

3. The fact that after HCV eradication data are 3-5 years post-DAA treatment should be mentioned as drawback of the study. Morevoer, a second analysis should be performed regarding Bil changes over time

4. Half of your patients were cirrhotics. An analysis should be made on whether the presence of cirrhosis was a factor regarding improvement of cholestasis

Comments on the Quality of English Language

Engliash language adequate- minor changes required

Author Response

We would like to thank the Reviewer for their thorough and accurate revision and for their comments. Here follows a point-by-point response to the comments:

- In introduction section, it should be mentioned that cryoglobulinemia is only one of the extrahepatic manifestations of HCV and that DAAs seem to have incosistent effects in them (ref: Kuna L, J Clin Transl Hepatol. 2019; Pol S, Nat Rev Gastroenterol Hepatol. 2018; Androutsakos T, Viruses 2023; Mazzaro C, Viruses 2021 among others)

We added a sentence in the introduction section to clarify that, as for cryoglobulinaemia, the use DAAs and other antiviral therapies not always are effective on extrahepatic manifestations of CHC. We added some of the references you kindly suggested.

- The 3 patients with HIV co-infection should be excluded, since both cholestasis and plasma cell infiltrates could be attributed to HIV infection per se (ref: Te HS, Clin Liv Dis 2004)

We thank the reviewer for its comment, we removed HCV/HIV coinfected patients and revised the statistical analysis.

- The fact that after HCV eradication data are 3-5 years post-DAA treatment should be mentioned as drawback of the study. Morevoer, a second analysis should be performed regarding Bil changes over time

This issue is definitely a limitation of our study and we added it to the discussion section. Unfortunately, follow-up bioumoral data for every patient were not taken at a fixed time interval, therefore to conduct the analysis on cholestasis after DAAs therapies, we relied on the most recent value obtained for each patient in the 3-5 year follow-up period.

-Half of your patients were cirrhotics. An analysis should be made on whether the presence of cirrhosis was a factor regarding improvement of cholestasis

The number of chirrotic patients did not significantly differ between the two groups (p>0,05) and there was no significant correlation at multivariate analysis between cholestasis and the presence of cirrhosis.

Reviewer 2 Report

Comments and Suggestions for Authors

To thank the authors for their interesting research on the association of cholestasis in patients with CM and HCM. 

The introduction is well written, clear and concise, and introduces the aim of the study in detail.

Only comment that in the second paragraph, in lines 41-42, modify that sentence that is not very understandable because concepts are repeated. 

In line 52 there is the acronym CHC which I imagine is chronic hepatitis C, write it before placing the acronym.

Place the materials and methods section after the introduction, then the results and finally the discussion. 

In the discussion add a comment on the lower percentage of steatosis in the MC+ group compared to MC+.

Also add a hypothesis or comment on the possible immunologic mechanism involved in the association of cholestasis and HCC with MC+.

Comments on the Quality of English Language

 Minor editing of English language required

Author Response

We would like to thank the reviewer for its comments and suggestions. Here is a point-by-point response to the Reviewer comments. 

- The introduction is well written, clear and concise, and introduces the aim of the study in detail. Only comment that in the second paragraph, in lines 41-42, modify that sentence that is not very understandable because concepts are repeated.

Thank you for your suggestion. The sentence at lines 41-42 has been modified to be less redundant.

- In line 52 there is the acronym CHC which I imagine is chronic hepatitis C, write it before placing the acronym.

Thank you for pointing it out. We wrote the full name before the acronym.

- Place the materials and methods section after the introduction, then the results and finally the discussion.

We followed the order of the sections as provided by the Journal’s manuscript form.

- In the discussion add a comment on the lower percentage of steatosis in the MC+ group compared to MC+.

As reported in Table 1, there were no significant differences in the number patients with steatosis between MC+ and MC- groups.

- Also add a hypothesis or comment on the possible immunologic mechanism involved in the association of cholestasis and HCC with MC++.

Thank you for your suggestion, we expanded in the discussion section the paragraph in which we hypothesize the possible cryoglobulins mediated mechanism of liver damage.

Round 2

Reviewer 1 Report

Comments and Suggestions for Authors

The manuscript is now greatly improved and, in my opinion, fit for publication

Comments on the Quality of English Language

Minor editing required